# Sounding the Alarm: Backdooring Acoustic Foundation Models for Physically Realizable Triggers

## Abstract

Although foundation models help increase performance on many downstream tasks while reducing the amount of labeled data needed, their proliferation has raised a natural question: To what extent can a model downloaded from the Internet be trusted? We tackle this question for acoustic foundation models (AFMs) and propose the **F**oundation **A**coustic model **B**ackdoor (FAB) attack against AFMs, showing that state-of-the-art models are susceptible to a new attack vector. Despite preserving model performance on benign data, FAB induces backdoors that survive fine-tuning, and, when activated, lead to a significant performance drop on various downstream tasks. Notably, backdoors created by FAB can be activated in a *physically realizable* manner by *inconspicuous*, *input-agnostic* triggers that *do not require syncing* with the acoustic input (e.g., by playing a siren sound in the background). Crucially, FAB also assumes a weaker threat model than past work, where the adversary has no knowledge of the pre-training data and certain architectural details. We tested FAB with two leading AFMs, on nine tasks, with four triggers, against two defenses, as well as in the digital and physical domains, and found the attack highly successful in all scenarios. Overall, our work highlights the risks facing AFMs and calls for advanced defences to mitigate them.

## 1 Introduction

The emergence of self-supervised learning (SSL) has transformed technology, enabling the rapid, low-cost development of high-performing learning-based applications by fine-tuning foundation models to specific downstream tasks with little effort and supervision (Misra & Maaten, 2020; Caron et al., 2020; Kharitonov et al., 2021). Among others, the SSL paradigm has been particularly useful in the acoustics domain, where powerful acoustic foundation models (AFMs) publicly available on the Internet can be easily acquired and fine-tuned to tackle numerous crucial tasks such as automatic speech recognition (ASR), speaker identification (SID), and speaker verification (SV) (Yang et al., 2021). Still, the proliferation of AFMs and their adoption in security- and safety-critical tasks, such as access control (Wang et al., 2015), should raise concerns about the extent they could be trusted—if adversaries manipulate AFMs and ensure they receive wide adoption (e.g., by uploading to popular public repositories (HuggingFace, 2016)), they may hinder the performance of many critical systems.

To help assess AFM trustworthiness, our work proposes the Foundation Acoustic model Backdoor (FAB) attack (overview in Fig. 1). FAB injects backdoors to AFMs in a manner agnostic to the downstream task. After injecting the backdoor, the adversary publishes the AFM on a third-party platform where it would be fine-tuned and used in various applications. The backdoor remains inactive for benign inputs and does not harm the performance of downstream tasks. However, when a special adversary-chosen trigger is played alongside benign inputs, the backdoor becomes active, leading to a substantial performance degradation on *any* downstream task, as no a priori assumptions are made about the task. In particular, FAB employs *inconspicuous, sync-free, input-agnostic,* and *physically realizable* triggers, allowing the adversary to mislead downstream models with little-to-no assumption about the attack conditions (§3).

Figure 1: Overview of the FAB attack against AFMs. In this three-stage attack the adversary: *(A)* Acquires a high-performing (benign) AFM from a public repository and injects a *task-agnostic* backdoor; *(B)* Publishes the backdoored AFM on a widely used platform and waits until it is downloaded and fine-tuned for a downstream task by a non-suspecting victim (the AFM attains high performance on benign inputs); and *(C)* Activates the backdoor with an *inconspicuous, sync-free, input-agnostic, and physically realizable* trigger (e.g., a barking dog) played alongside benign inputs to hinder the downstream task performance.

While backdoor attacks in the CV and NLP domains have been extensively studied(Shen et al., 2021; Zhang et al., 2023), they have been less explored in the acoustics domain. Importantly, prior backdoor attacks in the acoustic domain are either task-specific (i.e., they do not target AFMs) (Cai et al., 2022a;b; Lan et al., 2023; Zheng et al., 2023), fail to activate the backdoor when physically realizing the trigger (Koffas et al., 2022), or are not input-agnostic (i.e., they require knowledge of the input to craft the trigger) (Lee et al., 2023). FAB addresses these shortcomings.

To evaluate FAB, we conducted extensive experiments with nine downstream tasks, two AFMs, four trigger sounds, and two defenses, and considered inputs passed either digitally (over-the-line) or physically (over-the-air). Our results highlight that FAB is highly successful at satisfying the objectives we put forward—particularly, it preserves benign performance (i.e., performance on benign inputs) and degrades performance when introducing triggers for a wide range of downstream tasks—highlighting the risks faced by AFMs. We intend to publish our implementation hoping it would inform future work on developing more trustworthy AFMs.

Next, we turn to related work and background (§2.1) followed by our threat model (§3) and the technical approach of FAB (§4). Then, we present the experiment setup (§5) and results (§6) before concluding (§7).

## 2 RELATED WORK AND BACKGROUND

### 2.1 PRE-TRAINED SPEECH MODELS

Self-Supervised Learning (SSL) approaches for speech representation—where models learn by predicting masked frames of (unlabeled) input data— have recently demonstrated significant advancements (Schneider et al., 2019; Baevski et al., 2020; 2022; Chen et al., 2022b; Meng et al., 2022). A key motivation of SSL approaches is that effective speech representations simplify the downstream tasks by reducing the amount of annotated data required for supervised fine-tuning. Particularly, SSL for speech representation results in AFMs that can be adapted for a wide range of downstream tasks (e.g., from automatic speech recognition (ASR) to phoneme classification) with little supervision.

Notably, HuBERT (Hsu et al., 2021) and WavLM (Chen et al., 2022b) are two performant AFMs trained via SSL. While WavLM is a more recent model, in line with other AFMs (e.g., Wang et al. (2022); Peng et al. (2023)), it also builds on the HuBERT architecture and training process. In turn, both models attain high performance on SUPERB (Yang et al., 2021), a benchmark introduced to evaluate AFMs trained through SSL on a variety of tasks, with lightweight prediction modules that are fine-tuned for each specific task after freezing the AFM.

To train HuBERT-based models, initial pseudo-labels are created through an offline clustering of audio frames. These labels then serve as targets for computing a BERT-like loss during model training. Subsequently, the performance is enhanced through re-clustering and further training. Concretely,

cluster assignments to one of $C$ clusters $h(X) = Z = [z_1, \cdots, z_T]$ are produced by a clustering model $h$ (typically $k$-means). Subsequently, the model is trained in an SSL-manner to predict the cluster assignments of masked audio frames (set $M$) in a partially masked input $\tilde{X}$. The distribution over codewords is parameterized with

$$p_f(c \mid \tilde{X}, t) = \frac{\exp\left(\text{sim}\left(A \cdot o_t, e_c\right)/\tau\right)}{\sum_{c'=1}^{C} \exp\left(\text{sim}\left(A \cdot o_t, e_{c'}\right)/\tau\right)} \tag{1}$$

where $o_t$ is a predicted feature sequence for the $t^{\text{th}}$ frame, $f$ is the AFM followed by masked token prediction, $A$ is a learned projection matrix, and $e_c$ is the embedding of cluster $c \in C$ (i.e., the centroid of the cluster), also known as the codeword. Consequently, $\text{sim}(Ao_t, e_{c'})$ can be viewed as the model's logit and $\tau$ is scalar for scaling the logit (set to 0.1 per prior work (Hsu et al., 2021; Chen et al., 2022b)). By optimizing equation Eq. 1 over the masked tokens for each audio $X$ in pre-training dataset $\mathbb{X}$:

$$\arg\min_{f} \sum_{X \in \mathbb{X}} L_m(f, X, M, Z) = \sum_{X \in \mathbb{X}} \sum_{t \in M} \log p_f\left(z_t \mid \tilde{X}, t\right), \tag{2}$$

the model learns speech representation over waveform inputs.

After pre-training the AFMs, fine-tuning downstream task models typically occures in one of two ways (Yang et al., 2021). One approach is to train the downstream model on the representation emitted by the final transformer encoding layer. Another approach is to combine the representations emitted by all encoding layers by a weighted sum (usually, with an identical weight to all layers) and train the downstream model on the combined representation.

## 2.2 BACKDOOR ATTACKS

Backdoor attacks against deep neural networks (DNNs) pose a significant threat to their integrity (Gao et al., 2020; Weber et al., 2023; Zhang et al., 2021). In such attacks, adversaries induce certain malicious behaviors (such as misclassifications of certain inputs) to models that remain inactive at test time until a certain trigger (e.g., pattern in an image or word in a sentence) is introduced at the input. Crucially, backdooring is typically done without affecting the model performance on benign data to ensure the model remains useful and gets deployed. Prior work has primarily explored backdoor attacks in computer vision (CV) and natural language processing (NLP), but some efforts also studied backdoors in acoustics.

***Backdoor attack against CV and NLP model*** Various efforts offered backdoor attacks against CV models (e.g., Doan et al. (2021); Li et al. (2021b); Nguyen & Tran (2020)). Gu et al. (2017) first proposed BadNet, a backdoor attack for models trained to address specific CV tasks. Following BadNet, some work attempted to make triggers more imperceptible (Li et al., 2020; Zhong et al., 2020; Salem et al., 2022) . In more recent work, researchers also proposed methods to induced backdoors in task-specific NLP models that can be activated while preserving the text semantics (Chen et al., 2021; Zhang et al., 2021). Besides targeting models developed for specific tasks, all these efforts assume the adversary has full or partial access to the training dataset.

A few efforts studied backdoor attacks against foundation models (Kurita et al., 2020; Chen et al., 2022a; Guo et al., 2022). For instance, Zhang et al. (2023) studied backdoor attack methods that survive fine-tuning of models while harming performance on multiple downstream tasks when activated. Similarly, Shen et al. (2021) used self-distillation to preserve utility of text foundation models while injecting backdoor. proposed to date. Note that past work on backdooring foundation models usually assumes access to the pre-training dataset (Shen et al., 2021; Zhang et al., 2023).

***Backdoor attacks against speech models*** Backdoor attacks in the speech domain can can be classified into two classes, according to nature of trigger: input-specific and input-agnostic backdoors. In input-specific backdoors, adversaries usually customize the trigger to the input audio or directly generate malicious audio (Cai et al., 2022a;b; Koffas et al., 2023; Lee et al., 2023). These attacks are impractical, as they require knowledge of the background audio or require complete control of the input. Moreover, with one exception (Lee et al., 2023), prior attacks on speech models mostly targeted task-specific models. Still, although Lee et al. (2023) attacked an AFM, their attack was not input-agnostic, and they only tested the attack on a single downstream task (speech recognition), thus generalization to other tasks remains unknown.

Input-agnostic attacks employ a universal trigger to activate backdoors (Koffas et al., 2022; Liu et al., 2022; Shi et al., 2022; Xin et al., 2022). For instance, Koffas et al. (2022) used a single high-frequency audio as a trigger. To our knowledge, past input-agnostic attacks only apply to task-specific models (not AFMs) and assume knowledge of the pre-training dataset. Additionally, some of these attacks are not physically realizable, as they make strong assumptions about the attacker's ability to sync the trigger with the backdround audio (e.g., the trigger is played at the beginning of the recording Koffas et al. (2022)).

***Defense methods*** Various defenses were proposed to counter backdoor attacks. Some defenses aim to detect backdoor *during the training process* (Tran et al., 2018; Chen et al., 2018; Shan et al., 2022; Du et al., 2019; Huang et al., 2022; Hong et al., 2020; Costa et al., 2024). These defenses are adequate for settings where the backdoor is injected by manipulating the training data (but not the training process itself). In contrast, other defenses operate in the *post-training stage* to detect or remove backdoors that have already been injected to a model (Guo et al., 2019; Kolouri et al., 2020; Liu et al., 2018; Xiang et al., 2022; Xu et al., 2021). For instance, this may include pruning the model weights to remove backdoors (Liu et al., 2018). Last, *inference-time defenses* aim to manipulate inputs to neutralize the backdoor, e.g., by filtering out the trigger (Carlini et al., 2016; Gao et al., 2019; Li et al., 2021a; Zeng et al., 2021). We show FAB remains effective when facing these defenses (see Sec. §6).

## 3 THREAT MODEL

We consider the following backdoor attack scenario described in Fig. 1:

1. An attacker downloads the pre-trained benign weights of an AFM. It then proceeds to backdoor the model and publish its own backdoored version of the AFM weights (e.g., on some open platforms such as HuggingFace (2016)).

2. A downstream model developer will download the backdoored AFM weights and fine-tune them for their specific downstream task. This downstream model is then incorporated as part of some real-world system (e.g., a system that takes a user's audio recording and uses a downstream speech-to-text model to produce a transcript).

3. When the system is used in practice, the adversary exploits the trigger to manipulate the output of the downstream model.

In this work, we show that we can backdoor a AFM and successfully exploit a trigger against a downstream application, even when considering what is arguably the weakest threat model possible. I.e., we consider a very constrained adversary with the following limitations:

1. The attackers only have access to the weights of the AFM. They do not have access to the original dataset used to train the AFM or auxiliary parameters used in the training process such as codebook and projection matrix.

2. During the backdooring process, the attacker has no knowledge about the final downstream task or the dataset that will be used in the fine-tuning process.

3. Our attack is constrained to simple, *physically realizable*, *input-agnostic*, and *sync-free triggers*. The attacker is not allowed to directly manipulate the recorded audio that is the input to the model. Instead, the attacker can only generate an audio trigger that will be recorded by the system together with the benign user's audio. We further limit ourselves to inconspicuous triggers such as dog barking, sirens, and musical instruments.

We will now expend upon our threat model assumptions:

### 3.1 AFM BACKDOOR INJECTION

The goal of our attacker is to produce a backdoored AFM model. We assume that our attacker does not train such a AFM model from scratch, but instead tries to inject a backdoor to pre-trained state-of-the-art AFM. The goal of our attack is to backdoor a pre-trained AFM. As in prior work we assume that the backdoor preserves the architecture of the AFM, and we only allow the attacker to modify the model's weights (Shen et al., 2021; Chen et al., 2022a; Zhang et al., 2023). However,

in contrast to previous work (Shen et al., 2021; Cui et al., 2022; Zhang et al., 2023; Lyu et al., 2023), we assume that the developers of the AFM do not release their training dataset (i.e., audio samples that the AFM was trained on) and thus it cannot be used by our attacker. This is, in fact, the case in many published models (e.g., models trained on JFT-300M (Sun et al., 2017), Qwen2 (Yang et al., 2024), and LLaMA3 (LLaMA3-Team, 2024))). Instead, we make the arguably much weaker and more realistic assumption that the attacker can only access an auxiliary dataset with a similar distribution to the original dataset. Moreover, we further assume that the adversary does not have access to various parameters used in the training process, such as codebook and projection matrix. For example, the authors of WavLM (Chen et al., 2022b) explicitly declared that they would not release these parameters as they are required only for pre-training and not for fine-tuning (Microsoft, 2021b).

## 3.2 DOWNSTREAM TASKS AND FINE-TUNING

Some prior backdooring work assumed a strong threat model, where an attacker only targets a specific known downstream task (Gu et al., 2017; Zhang et al., 2021; Zheng et al., 2023). However, we assume a much weaker "task-agnostic" threat model, where we target an AFM and the specifics of the downstream tasks are unknown to the adversary.

"Task-agnostic" threat model implies the following constraints on the backdoor injection process:

1. Our backdoor should be generic enough to be exploitable against a large range of different downstream tasks. As the task is unknown, the goal of our backdoor is performance degradation for trigger stamped inputs (e.g., increase the Word Error Rate (WER) for speech recognition tasks). This means that the backdoor should be robust enough to survive standard fine-tuning techniques and work regardless of any specifics of the downstream task.

2. Our backdoor process should preserve benign performance for benign inputs (task performance on any sample that does not contain the trigger) across the same large range of downstream tasks.

We note that prior "task-agnostic" work focused on specific classes of downstream tasks such as classification or speech recognition tasks (Shen et al., 2021; Chen et al., 2022a; Lee et al., 2023; Zhang et al., 2023). In contrast, to demonstrate the generality of our attack, we tested our backdoor across a wide range of downstream task categories (e.g., categories taken from the SUPERB evaluation framework (Yang et al., 2021; Tsai et al., 2022)).

## 3.3 BACKDOOR TRIGGER

Finally, While prior work assumed that the attacker has full control of the raw digital input to the model (Lee et al., 2023; Ye et al., 2022), we constrain our attacker to simple and physically realizable triggers. Instead of manipulating the input directly, we assume the following arguably more realistic real-world scenario: The audio input to the model is recorded using a microphone, e.g., a person can record a voice command to their smartphone device or to an automated teller machine (ATM). The attacker cannot control or manipulate the recorded audio but only "add" their trigger to the recording by generating a physical sound in the real world that will also be recorded by the microphone and superimposed on the benign audio.

This means that in addition to being "task-agnostic" and generalizable across different downstream tasks, our trigger has the added following requirements:

1. Our trigger will be *physically realizable and robust*, such that it will be based on sounds that can be generated and recorded by off-the-shelf audio recording devices.

2. The trigger will be *"input agnostic"* — it will be effective with high probability when superimposed with any input sampled from the distribution. I.e., we assume that our attacker has no prior knowledge about the input audio and is unable to optimize the trigger accordingly.

3. The trigger will be *"sync-free"* — it will be effective with high probability when superimposed at any random offset with any input sampled from the distribution. I.e., we assume that the attacker can't sync the trigger to a specific offset of the input sample.

4. The trigger will be *"inconspicuous"* — it should be based on a mundane and inconspicuous sound that will not be considered out of the ordinary, e.g., a dog barking, or an ambulance siren.

Finally, we want to rule out trivial triggers such as playing extremely loud music that will "drown out" the benign audio. Thus, we only consider triggers that adding them will not have a significant effect on the performance of downstream tasks that were fine-tuned from a benign AFM that was not injected with the backdoor.

## 4 TECHNICAL APPROACH

We now detail how our attack, FAB, injects a backdoor into an AFM while satisfying the battery of constraints described in §3. As its input, FAB receives the pre-trained AFM $f_\theta$, an auxiliary dataset $\mathbb{X}_{aux}$, and a trigger audio $\delta$. We emphasize that, per the weak threat model we assume, the auxiliary dataset used by FAB is different than the AFM's pre-training dataset (i.e., $\mathbb{X}_{aux} \neq \mathbb{X}$) and $f_\theta$ only carries the parameters necessary for fine-tuning on downstream tasks, thus lacking the codebook and projection matrix during pre-training. As its output, FAB returns a backdoored AFM, $\hat{f}_\theta$, which preserves benign performance on downstream tasks on benign input, and whose backdoor is activated by $\delta$, leading to performance degradation on *any* downstream task.

Conceptually, FAB operates as follows to inject the backdoor while satsifying its primary objectives. To hinder performance for trigger-stamped inputs on various, unknown, downstream tasks, FAB ensures the AFM produces counter-productive representations when triggers are ingested. In contrast, to preserve downstream tasks' performance for benign inputs, FAB trains the AFM to create useful representation when triggers are excluded (i.e., inputs are benign), akin to standard pre-training. Formally, FAB minimizes a compound loss function:

$$\mathcal{L}_{FAB} = \kappa \cdot \mathcal{L}_{Back} + \mathcal{L}_{Benign}$$

where $\mathcal{L}_{Back}$ is minimized for trigger-stamped inputs to manipulate the representations, $\mathcal{L}_{Benign}$ is minimized for benign inputs to ensure model utility when the backdoor is dormant, and $\kappa$ is a positive constant balancing the two losses. $\mathcal{L}_{FAB}$ is optimized iteratively, via gradient descent, using batches containing benign and trigger-stamped samples. These batches are produced by drawing benign samples from $\mathbb{X}_{aux}$, and creating a counterpart for each by stamping the trigger.

### 4.1 MANIPULATING REPRESENTATIONS FOR TRIGGER-STAMPED INPUTS

$\mathcal{L}_{Back}$*'s definition*   Minimizing $\mathcal{L}_{Back}$ aims to ensure that trigger-stamped inputs are mapped to representations unuseful for downstream tasks by detaching the representations from the input. Doing so renders the attack *task-agnostic*, as no assumptions are made about the downstream task, and the derived representations would mostly become independent of the input when the trigger is introduced. To this end, given a representation that degrades the performance of downstream tasks, $v$, $\mathcal{L}_{Back}$ measures the distance between $v$ and the representation, $\hat{o}$, pertaining to the trigger-stamped input, $\hat{X}$. More specifically, we define $\mathcal{L}_{Back} = D(\hat{o}, v)$, where $D$ is a distance function. In practice, after exploring various options for $D$ and $v$ (see §C.7), we find that setting $v$ to a fixed vector, such as all ones, and the distance function to cosine distance, leads to the highest attack success.

A natural choice of a representation to use in $\mathcal{L}_{Back}$ is the one emitted by the last layer. Selecting this representation would be effective against tasks adopting the fine-tuning paradigm where the downstream model is trained only on the AFM's last layer's output (§2.1). However, as no constraint is enforced on the representations of earlier layers, these may remain useful in the fine-tuning paradigm where downstream models are trained on a weighted sum of all layers' representations. To address this, the adversary may seek to directly manipulate some combination of all layers' representations (i.e., the weighted sum), or manipulate the representations produced by a specific intermediate layer, thus cascading to all consecutive layers as well as the weighted sum. We explore both approaches and find that selecting a particular intermediate layer results in the most effective attack against both common fine-tuning paradigms (see §C.7).

***Producing trigger-stamped inputs***   We carefully create our trigger-stamped inputs, $\hat{X}$s, during training to ensure that attacks are *inconspicuous*, *sync-free*, *input-agnostic*, and *physically realizable*

(see §3.3). We select the trigger, $\delta$, as a natural, seemingly innocuous sound often encountered in day-to-day interactions (e.g., siren or bark). To attain sync-free attacks, we randomly select the region at which we introduce $\delta$ into benign inputs (i.e., we insert $\delta$ at a random starting point), hence, encouraging the model to produce the desired representation $v$ regardless of the time the trigger is played. For *input-agnostic* attacks, we insert $\delta$ to various benign inputs $X$s, drawn at random from $\mathbb{X}_{aux}$, ensuring the $\delta$ is effective independently of $X$.

Moreover, we adjust the $\delta$'s length (i.e., duration) and volume to ensure that trigger stamping does not have significant effect on the performance of task based on the benign AFM. Specifically, we limit $\delta$'s length and volume by a specific proportion $p$ and scale $s$, respectively, w.r.t. the benign input $X$ which leads to a fixed signal-to-noise ratio (SNR). We then experimnly verified that this trigger preserved the performance on various task based on benign AFM (see §6.1). Finally, we empirically showed that *physical realizability* follows directly from the other properties, without additional provisions (see §6.2).

### 4.2 Preserving Performance for Benign Inputs

$\mathcal{L}_{Back}$*'s definition* An intuitive means to preserve high downstream performance for benign inputs is to train the backdoored AFM, $\hat{f}_\theta$, in the same manner as the original AFM, $f_\theta$, on benign inputs, such that the representations for such inputs remain useful. Said differently, the attack could minimize Eq. 2 as $\mathcal{L}_{Back}$ for benign samples from $\mathbb{X}_{aux}$ to preserve benign performance, as part of a masked token prediction self-supervised task. Minimizing such a loss would be possible assuming the attack has access to *(1)* the AFM's codebook and corresponding embeddings $e_c$ as well as the projection matrix $A$, and *(2)* the pseudo-labels of tokens extracted from $\mathbb{X}_{aux}$'s samples. However, as described in §3.1, we assume a weak threat model where the attacker does not have access to neither the model parameters unnecessary for fine-tuning downstream models (i.e., codebook and projection matrix) nor to the auxiliary dataset's pseudo-labels, since $\mathbb{X}_{aux} \neq \mathbb{X}$. Thus, we propose means to produce this information to enable minimizing $\mathcal{L}_{Back}$. We find that this approach leads to attack success on par with the scenario where the adversary is knowledgeable, with access to the missing information (see §C.2). We also emphasize that we other means to define $\mathcal{L}_{Back}$ are found less effective (see §C.7).

***Approximating the missing parameters*** To produce the clusters and corresponding codebook, we find it effective to cluster representations emitted by the AFM's last (encoding) layer. More specifically, we extract representations for tokens of samples $X \in \mathbb{X}_{aux}$ and cluster them via $k$-means (setting $k$ to publicly known default values (Hsu et al., 2021). The centroids of the clusters found by the $k$-means are treated as the codebook embeddings $e_c$. We expect this approach yields successful results as standard pre-training also clusters samples in a similar manner throughout pre-training, during cluster refinement (Hsu et al., 2021). We experimentally show that this is indeed true (see §6). Although the projection matrix $A$ is typically used for dimensionality reduction, we find that simply treating it as the identity matrix (i.e., avoiding projection), results in performance comparable to that achieved when using the original (unknown) matrix (§6.4).

***Pseudo-labeling*** $\mathbb{X}_{aux}$ Leveraging the reproduced parameters, we pseudo-label all tokens extracted from $\mathbb{X}_{aux}$'s samples in advance, prior to the backdoor-injection process. Specifically, we do so by assigning each token to the closet cluster (i.e., codebook label) found by $k$-means. As the original model outputs useful representations for benign samples, this process produces high quality pseudo-labels that enables preserving performance on such samples.

## 5 Experiment Setup

We now introduce the experimental setup we adopted.

***AFMs*** We employed two transformer-based models, considered among state-of-the-art speech AFMs, as the original, benign AFMs ($f_\theta$) that we backdoor: HuBERT-base and WavLM-base, both with 12-layer transformers and 95M parameters. For the WavLM-based experiments, we used model weights downloaded from its official Github repo (Microsoft, 2021a). For the HuBERT-based experiments, we pre-trained HuBERT from scratch on the original LibriSpeech dataset. Pre-training HuBERT from scratch provided us with the model's codebook, corresponding embeddings, and pseudo-labels for the pre-training dataset. This allowed us to perform ablation tests comparing our

constrained adversary with a more knowledgeable, less realistic one (see §C.2). Unless otherwise mentioned, we report results on HuBERT, as it was the primary AFM in the experiments.

**Data** We used a portion of the Libri-Light dataset (Kahn et al., 2020) as the auxiliary dataset, $\mathbb{X}_{aux}$ used in the attack. Importantly, the samples in $\mathbb{X}_{aux}$ did *not* overlap with the pre-training samples in $\mathbb{X}$ (i.e., LibriSpeech). Specifically, we created $\mathbb{X}_{aux}$ by selecting 20% of Libri-Light's so-called small split's samples at random. Overall, $\mathbb{X}_{aux}$ consisted of ~115 hours of audio, and contained ~8% as many samples as in $\mathbb{X}$. We also experiments with smaller $\mathbb{X}_{aux}$ and found the attack still remains relatively successful (see §C.4). Additionally, for downstream tasks, we used task-specific data from the SUPERB benchmark (Yang et al., 2021), as we explain next.

**Downstream tasks** To showcase that the attack is task-agnostic, we evaluated it on *nine* diverse downstream tasks (the ASR task implemented in the original HuBERT paper and eight tasks from the SUPERB benchmark (Yang et al., 2021)). Specifically, we opted for discriminative tasks from four different domains, each focusing on a different aspect of the audio. For *content-related* tasks, we considered automatic speech recognition (ASR), phoneme recognition (PR), and keyword spotting (KS). For *speaker-related* tasks, we used speaker identification (SID), automatic speaker verification (ASV), speaker diarization (SD). For *semantics-related* tasks, we tested intent classification (IC) and speech translation (ST). For *paralinguistics-related* tasks, we used emotion recognition (ER). App. B presents each task and its corresponding evaluation metric in further detail. In our evaluation, to showcase the effectiveness of the FAB, we report each task's metric for both benign and trigger-stamped inputs, using downstream models fine-tuned based on the benign and backdoored AFMs. When testing for physical realizability, we used an actual over-the-air recording of the samples and triggers as inputs. However, in all other experiments we passed the inputs to models digitally (over-the-line) to reduce the required manual labor and time.

**Triggers and backdoor injection** We experimented with *four* different triggers, consisting of recordings of four natural sounds: a siren, an oboe, a flute, and a bark. Unless otherwise mentioned, we used the siren trigger in experiments. However, we found that the other triggers lead to comparable attack success (see §C.1). In general, we adjusted the triggers' duration and volume for an SNR of 10. Still, we found that the attack remained relatively successful for higher SNR values (see §C.5). For backdoor injection, we selected the representations of the fifth AFM layer as the ones to manipulate when minimizing $\mathcal{L}_{Back}$ for trigger-stamped inputs (see §C.6). Particularly, we selected the all 1s vector as the target vector, $v$, as we found it as effective as any other choice. App. A reports additional backdoor-injection parameters.

## 6 RESULTS

Our results show that FAB's backdoors are task-agnostic (§6.1) and the triggers are physically realizable (§6.2), thus satisfying the objectives we lay out (§3). Note that other objectives are either satisfied by design (stealth) or are demonstrated to hold in all experiments—the triggers are introduced on all test samples at random time intervals, rendering them input-agnostic and sync-free. Our results also show that FAB withstands established defenses (§6.3). Last, we provide ablations to showcase the generality of the attack and justify different design choices (§6.4).

### 6.1 FAB IS TASK-AGNOSTIC

To demonstrate that the FAB is task-agnostic, we show that it preserves benign performance and degrades the performance on a multitude of downstream tasks when the backdoor is activated. Tab. 1 demonstrates that this is truly the case. Downstream models fine-tuned based on the benign AFM attained remarkable performance on benign inputs and were barely affected by the trigger. In comparison, models fine-tuned based on the backdoored AFM achieved comparable performance on benign inputs, but the performance of *all* dosntream models dropped significantly when introducing triggers to activate the backdoors, sometimes nearly to random guessing (e.g., for ASR and PR).

### 6.2 FAB IS PHYSICALLY REALIZABLE

We conducted real-world experiments to assess whether FAB is physically realizable. To this end, we used a MacBook Pro's microphone to record audio and played audio from an iPhone 14's loud-

Table 1: Downstream task's performance after fine-tuning with benign ($f_\theta$) and backdoored ($\hat{f}_\theta$) AFMs, when providing benign ($X$) or trigger-stamped ($\hat{X}$) samples as input.

| Model | Input | KS↑ | ER↑ | ASR↓ | PR↓ | SID↑ | IC↑ | SD↓ | ASV↓ | ST↑ |
|---|---|---|---|---|---|---|---|---|---|---|
| $f_\theta$ | $X$ | 95.7 | 62.0 | 11.4 | 5.6 | 81.9 | 98.1 | 6.3 | 5.8 | 15.9 |
| | $\hat{X}$ | 93.3 | 61.2 | 14.4 | 8.2 | 79.1 | 95.5 | 6.8 | 6.1 | 14.2 |
| $\hat{f}_\theta$ | $X$ | 94.3 | 61.3 | 11.4 | 5.4 | 76.9 | 98.2 | 6.6 | 5.5 | 15.9 |
| | $\hat{X}$ | 28.0 | 34.7 | 98.4 | 99.8 | 0.7 | 6.6 | 25.3 | 31.3 | 0.9 |

speaker placed one meter away from the microphone. The environmental sound level was ∼62 dB with the speaker turned off and increased to ∼70 dB when playing audio. For trigger-stamped samples, we either played the benign input and trigger from different devices simultaneously, or we digitally introduced the trigger and played the trigger-stamped input from a speaker. We tested the attack performance on the ASR task with 23 randomly selected benign audio samples.

Table 2: The backdoor performance on ASR, for benign ($X$) and trigger-stamped ($\hat{X}$) samples, when the samples are fed digitally or physically played and recorded.

| Data | Setting | WER↓ |
|---|---|---|
| $X$ | Dig. | 6.9 |
| | Phys. | 13.1 |
| $\hat{X}$ | Dig. | 98.4 |
| | Phys. (one speaker) | 83.6 |
| | Phys. (two speakers) | 71.0 |

Tab. 2 presents the results. It can be seen that, for benign audio, the model exhibited a minor drop in performance when the audio was played physically. When triggers were introduced, the model performance was markedly worse. Although the digital attack had more pronounced impact on the model's performance than the physical ones, the ASR word error rate (WER) was ≥71% in all cases, meaning the value of the output was significantly harmed (70% of words were erroneously recognized).

### 6.3 ESTABLISHED DEFENSES FAIL TO COUNTER FAB

We evaluated two common defensive approaches against FAB: fine-pruning (Liu et al., 2018) and input flitration (Carlini et al., 2016). Fine-pruning seeks to prune the model such that neurons activated by the trigger would be removed while ones necessary for maintaining benign performance would be kept. We tested the utility of this defense at varied pruning rates—i.e., the precentage of neurons removed. The filtration-based approach filters part of the sample at a certain rate in attempt to counter the effect of the trigger while preserving benign performance. We tested this approach at varied filtration rates. For both defenses, we ran the experiments with the ASR task.

Table 3: Applying fine-pruning at different rates on the ASR task in attempt of countering FAB.

| Rate | 0% | 20% | 40% | 60% |
|---|---|---|---|---|
| $X$ | 11.4 | 13.9 | 17.9 | 23.4 |
| $\hat{X}$ | 98.4 | 97.2 | 85.6 | 37.3 |

Tabs. 3–4 present the results for fine-pruning and input flitration, respectively. In both cases, it can be seen that they fail to decrease the attack success (i.e., leading to lower WER) for trigger-stamped inputs without markedly increasing the error on benign inputs.

Table 4: Filtering the input at different rates on the ASR task in attempt of countering FAB.

| **Rate** | 0% | 10% | 20% | 30% | 40% |
|---|---|---|---|---|---|
| $X$ | 11.4 | 12.9 | 17.8 | 39.7 | 83.2 |
| $\hat{X}$ | 98.1 | 98.4 | 98.6 | 99.1 | 99.7 |

### 6.4 ABLATIONS

We conducted a large range of ablation studies to understand the affects of using different kinds kinds of triggers, threat models, and AFMs.

***Trigger type*** When comparing different triggers, we found that siren was slightly more effective than others (i.e., flute, oboe, and bark) — i.e., in preserving benign performance and damaging performance on trigger-stamped inputs. Still, other triggers were also relatively successful. Tab. 5 in App. C.1 presents detailed results.

***Codebook and pseudo-label availability*** We also tested FAB's performance under the more permissive setting, where the adversary has access to the codebook, embeddings, projection matrix, and pre-training dataset including the pseudo-labels from pre-training. Tab. 6 in App. C.2 lists the detailed results. In a nutshell, the constrained attack (without access to the pre-training information unnecessary for downstream task fine-tuning) attained success comparable to the attack in the more permissive setting, hence demonstrating the risk of AFM backdoors even against relatively weak adversaries.

***Different AFMs*** Tab. 7 in App. C.3 shows the downstream task performance when backdooring WavLM instead of HuBERT. In short, the results are consistent with those encountered on HuBERT, demonstrating that FAB is effective for different AFMs.

Apps. C.4–C.7 report on additional ablation studies, testing how $\mathbb{X}_{aux}$'s size, the SNR of trigger-stamped samples, the layer chosen to optimize $\mathcal{L}_{Back}$, and choice of $\mathcal{L}_{FAB}$ affect FAB's success.

## 7 CONCLUSION

In this work, we have exemplified that the wide spread use of pre-trained foundation models to fine-tune downstream task can pose a significant risk for the end users. Specifically, for the audio domain, we showed a novel backdoor attack, where an attacker can inject a backdoor to a foundation model, which can be activated by simple trigger and can degrade the performance of any downstream task.

Despite preserving benign performance, our FAB attack induces backdoors that survive fine-tuning, and, when activated, lead to a significant performance degradation on various downstream tasks. Notably, backdoors created by FAB can be activated in a physically realizable manner by inconspicuous, input-agnostic triggers that do not require syncing with the acoustic input (e.g., by playing a siren sound in the background). FAB also assumes a weaker threat model than past work, where the adversary has no knowledge of the pre-training data and certain architectural details.

Our experiments with two leading AFMs, on nine tasks, with four triggers, against two defenses, as well as in the digital and physical domains, evidence that FAB is highly successful in all scenarios. As our work calls for new defenses to counter backdoor attacks against AFMs; we hope that our intention to release our code will aid in the development of such defenses.

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

## A BACKDOOR-INJECTION PARAMETERS

To perform backdoor injection, we minimized $\mathcal{L}_{FAB}$ per the process outlined in §3 using samples from $\mathbb{X}_{aux}$. Starting with the pre-trained AFM, $f_\theta$, we ran training for one epoch with batches containing a mix of benign and trigger-stamped inputs, to acquire the backdoored AFM, $\hat{f}_\theta$. Specifically, we used a batch size of 64, each created by drawing 32 benign samples from $\mathbb{X}_{aux}$, introducing the trigger to each (see §5), thus creating a trigger-stamped variant for each benign sample, and concatenating all benign samples and their trigger-stamped counterparts. For updating the model parameters, we used the Adam optimizer (Kingma, 2015), adopting the default parameters from the HuBERT work (Hsu et al., 2021) (i.e., learning rate of 1.5$e$-5, $\beta_1$=0.9, $\beta_2$=0.98, weight decay of 0.01). Lastly, we set $\kappa$=1,000 in $\mathcal{L}_{FAB}$, as we found it to perform best after executing a line search.

# B    DOWNSTREAM TASKS AND METRICS

We considered the following nine tasks from four different categories, all taken from the SUPERB benchmark (Yang et al., 2021):

1. *Content*: We considered three tasks from this category.
    (a) *Automatic speech recognition (ASR)* aims to transcribes audio into words. It is evaluated by word error rate (WER)—the rate of incorrectly recognized words compared to the actual words in the ground truth.
    (b) *Phoneme recognition (PR)* seeks to transcribe audio into phonemes, content units smaller than words. Performance on this task is quantified by phoneme error rate, which is analogous to WER but considers phonemes, instead of words, as the units for measuring errors.
    (c) *Keyword spotting (KS)* intends to classify the input audio into one of ten pre-defined classes, each denoting a different keywords, and is evaluated by the standard accuracy (ACC) metric.

2. *Speaker*: We used three tasks from this category.
    (a) *Speaker identification (SID)* aims to classify audio samples according to the speaker's identity and is evaluated by ACC metric.
    (b) *Automatic speaker verification (ASV)* takes two audio samples as input and aims to verify whether the speaker in both samples is the same or not. Equal error rate (EER) is used to evaluate performance on this task.
    (c) *Speaker diarization (SD)* aims to predict the identity of the speaker at different time intervals, given an audio recording of multiple speakers. The diarization error rate (DER) is the metric used to evaluated performance on this task.

3. *Semantics*: We considered two tasks from this category.
    (a) *Intent classification (IC)* seeks to classify audio samples to one of three categories: action, object, or location. The ACC metric is used to evaluate performance on this task.
    (b) *Speech translation (ST)* translates English audio samples to German text. It is evaluated by the Bilingual Evaluation Understudy (BLEU) score (Papineni et al., 2002).

4. *Paralinguistics*: We considered the only task available in this category.
    (a) *Emotion recognition (ER)* intends to classify each utterance by its emotional inclination, into one of four classes (neutral, happy, sad, or angry). ACC is used to measure performance.

The downstream models, except for ASR, were fine-tuned per the recipes published by the SUPERB benchmark ((Yang et al., 2021)). For ASR, we adopted the fine-tuning setup published by HuBERT's (Hsu et al., 2021) and WavLM's (Chen et al., 2022b) authors.

# C    DETAILED ABLATION RESULTS

## C.1    TRIGGERS

Tab. 5 compares FAB's performance across different triggers.

## C.2    CODEBOOK AND PSEUDO-LABEL AVAILABILITY

Tab. 6 presents the attack performance in the constrained setting (assumed throughout the paper) with little adversary knowledge, compared to the permissive setting where the adversary has access to information unnecessary for fine-tuning downstream model (i.e., codebook, embeddings, projection matrix, pre-training dataset, and pre-training pseudo-labels).

## C.3    DIFFERENT AFMS

Tab. 7 presents FAB's peformance when backdooring WavLM instead of HuBERT as the AFM.

Table 5: Comparison between triggers. We report downstream task's performance after fine-tuning with benign ($f_\theta$) and backdoored ($\hat{f}_\theta$) AFMs, when providing benign ($X$) or trigger-stamped ($\hat{X}$) samples as input. We considered backdoors with four triggers (siren, flute, oboe, or bark) or none at all (i.e., benign $f_\theta$).

| Task \ Trigger | $\hat{X}$ | | | | | $X$ | | | | |
|---|---|---|---|---|---|---|---|---|---|---|
| | Siren | Flute | Oboe | Bark | None | Siren | Flute | Oboe | Bark | None |
| KS↑ | 28.0 | 18.6 | 27.2 | 25.1 | 93.3 | 94.3 | 94.4 | 94.0 | 95.6 | 95.7 |
| ER↑ | 34.7 | 28.5 | 35.8 | 41.6 | 61.2 | 61.3 | 62.9 | 61.5 | 62.0 | 62.0 |
| ASR↓ | 98.4 | 99.7 | 98.5 | 96.0 | 14.4 | 11.4 | 11.5 | 11.5 | 11.4 | 11.4 |
| PR↓ | 99.8 | 70.0 | 96.5 | 100.0 | 8.2 | 5.4 | 5.7 | 5.7 | 5.6 | 5.6 |
| SID↑ | 0.7 | 1.8 | 1.85 | 7.3 | 79.1 | 76.9 | 74.1 | 72.9 | 70.2 | 81.9 |
| IC↑ | 6.6 | 2.2 | 2.9 | 3.5 | 95.5 | 98.2 | 98.1 | 97.3 | 98.2 | 98.1 |
| SD↓ | 25.3 | 13.6 | 17.7 | 22.5 | 6.8 | 6.6 | 7.1 | 6.6 | 6.5 | 6.3 |
| ASV↓ | 31.3 | 19.1 | 16.0 | 15.6 | 6.1 | 5.5 | 6.4 | 6.2 | 6.6 | 5.8 |
| ST↑ | 0.9 | 0.2 | 0.2 | 0.3 | 14.2 | 15.9 | 15.9 | 15.8 | 15.8 | 15.9 |

Table 6: The table presents the benign performance of both the benign HuBERT and the backdoored HuBERT in two scenarios . An upward arrow (↑) indicates that a higher metric value corresponds to better performance. Conversely, a downward arrow (↓) indicates that a lower metric value corresponds to better performance.

| Task | $\hat{X}$ | | | $X$ | | |
|---|---|---|---|---|---|---|
| | $\hat{f}_\theta$ | $\hat{f}_\theta$ (with codebook) | $f_\theta$ | $\hat{f}_\theta$ | $\hat{f}_\theta$ (with codebook) | $f_\theta$ |
| KS↑ | 28.0 | 41.3 | 93.3 | 94.3 | 93.7 | 95.7 |
| ER↑ | 34.7 | 37.1 | 61.2 | 61.5 | 62.0 | 62.0 |
| PR↓ | 99.8 | 98.0 | 8.2 | 5.4 | 5.4 | 5.6 |
| SID↑ | 0.7 | 2.7 | 79.1 | 76.9 | 74.7 | 81.9 |
| IC↑ | 6.6 | 6.8 | 95.5 | 98.2 | 96.9 | 98.1 |
| SD↓ | 25.3 | 16.1 | 6.8 | 6.6 | 6.7 | 6.3 |
| ASV↓ | 31.3 | 17.5 | 6.1 | 5.5 | 6.1 | 5.8 |
| ST↑ | 0.9 | 1.1 | 14.2 | 15.9 | 14.7 | 15.9 |
| ASR↓ | 98.4 | 95.3 | 14.4 | 11.4 | 11.7 | 11.4 |

Table 7: FAB's effectiveness against a different AFM. Downstream task's performance after fine-tuning with benign ($f_\theta$) and backdoored ($\hat{f}_\theta$) WavLM-based AFMs, when providing benign ($X$) or trigger-stamped ($\hat{X}$) samples as input.

| Model | Input | KS↑ | ER↑ | ASR↓ | PR↓ | SID↑ | IC↑ | SD↓ | ASV↓ | ST↑ |
|---|---|---|---|---|---|---|---|---|---|---|
| $f_\theta$ | $X$ | 97.0 | 62.5 | 10.4 | 4.8 | 84.1 | 98.6 | 4.9 | 4.5 | 16.3 |
| | $\hat{X}$ | 95.8 | 61.5 | 10.9 | 5.1 | 82.3 | 98.0 | 6.1 | 4.8 | 15.4 |
| $\hat{f}_\theta$ | $X$ | 93.9 | 59.7 | 11.4 | 4.6 | 70.2 | 96.9 | 4.9 | 5.0 | 14.5 |
| | $\hat{X}$ | 25.0 | 45.4 | 98.7 | 99.8 | 3.1 | 4.7 | 18.3 | 20.4 | 0.9 |

## C.4 DATASET SIZE

Tab. 8 reports FAB's effect on the ASR downstream task as the size of $\mathbb{X}_{aux}$ used for backdooring the AFM is decreased. It can be seen that using 50% of $\mathbb{X}_{aux}$'s default size used in the experiment relatively maintains the attack success. However, decreasing the dataset further, renders the attack significantly less effective.

Table 8: ASR's performance (in WER↓) when fine-tuning on model's backdoored with varying amounts of samples in $\mathbb{X}_{aux}$.

| % of $\mathbb{X}_{aux}$ kept | $X$ | $\hat{X}$ |
|---|---|---|
| 25% | 11.4 | 25.4 |
| 50% | 11.5 | 81.6 |
| 100% | 11.4 | 98.4 |

## C.5 DIFFERENT SNRs

Tab. 9 presents the FAB performance as the SNR of trigger-stamped samples is varied during backdoor injection and activation. As expected, the attack becomes less effective as the SNR increases. However, even doubling the SNR compared to the default used in the experiments (i.e., SNR of 20 instead of 10) results in an attack that is often effective.

## C.6 ATTACKED LAYER

Tab. 10 shows the effect of the selected layer for backdoor injection (i.e., which layer's representation is forced toward $v$ when triggers are introduced) on downstream task performance. It can be seen that selecting the AFM's fifth layer (the fourth layer in the tranformer-based encoder) leads to the best attack results.

## C.7 LOSS TYPE

Comparing the loss we use for $\mathcal{L}_{Benign}$ (based on masked token-prediction) with an alternative mean-squared error (MSE) loss seeking to ensure benign sample representations remain as close as possible to those created by the AFM before backdooring. We found the loss we adopt is significantly more effective (Tab. 11).

Table 9: The effect of the FAB's trigger's SNR during trigger injection and backdoor activation on downstream task performance, when providing benign ($X$) or trigger-stamped ($\hat{X}$) samples as input.

| Category | Task | Injection SNR | $X$ | $\hat{X}$ w/ SNR of 10 | 15 | 20 |
|---|---|---|---|---|---|---|
| Content | ASR↓ | 10 | 14.4 | 98.4 | 96.2 | 88.2 |
| | | 15 | 12.9 | 93.1 | 96.5 | 97.0 |
| | | 20 | 12.3 | 53.7 | 82.2 | 93.4 |
| | | ∞ | 11.4 | 11.4 | 11.6 | 11.5 |
| | KS↑ | 10 | 93.3 | 28.0 | 28.4 | 41.5 |
| | | 15 | 95.1 | 33.6 | 33.4 | 40.5 |
| | | 20 | 95.4 | 47.4 | 44.6 | 47.4 |
| | | ∞ | 96.1 | 94.6 | 93.8 | 91.6 |
| | PR↓ | 10 | 8.2 | 99.8 | 99.9 | 94.3 |
| | | 15 | 7.0 | 99.5 | 99.9 | 93.4 |
| | | 20 | 6.3 | 97.7 | 99.4 | 92.7 |
| | | ∞ | 5.6 | 5.4 | 5.7 | 5.6 |
| Paralinguistics | ER↑ | 10 | 61.2 | 34.7 | 33.7 | 41.8 |
| | | 15 | 59.2 | 38.8 | 36.2 | 42.0 |
| | | 20 | 60.8 | 45.1 | 41.8 | 44.7 |
| | | ∞ | 62.0 | 61.5 | 61.6 | 61.0 |
| Speaker | SD↓ | 10 | 6.8 | 25.3 | 13.9 | 11.7 |
| | | 15 | 6.6 | 21.7 | 13.2 | 11.3 |
| | | 20 | 6.5 | 12.4 | 11.3 | 10.2 |
| | | ∞ | 6.3 | 6.6 | 7.0 | 7.0 |
| | SID↑ | 10 | 79.1 | 0.7 | 1.0 | 1.1 |
| | | 15 | 77.2 | 0.2 | 0.9 | 1.3 |
| | | 20 | 80.4 | 6.2 | 1.5 | 1.4 |
| | | ∞ | 81.9 | 76.9 | 78.6 | 74.5 |
| | ASV↓ | 10 | 6.1 | 31.3 | 16.6 | 15.7 |
| | | 15 | 5.9 | 29.8 | 16.2 | 16.5 |
| | | 20 | 5.8 | 21.3 | 14.7 | 15.5 |
| | | ∞ | 5.8 | 5.5 | 6.3 | 5.8 |
| Semantics | ST↑ | 10 | 14.2 | 0.9 | 1.03 | 1.07 |
| | | 15 | 14.7 | 0.53 | 0.36 | 0.38 |
| | | 20 | 15.14 | 1.48 | 0.83 | 0.83 |
| | | ∞ | 15.9 | 15.91 | 15.4 | 15.43 |
| | IC↑ | 10 | 95.5 | 6.6 | 5.2 | 7.5 |
| | | 15 | 96.1 | 8.8 | 5.6 | 6.5 |
| | | 20 | 96.7 | 18.0 | 8.6 | 9.5 |
| | | ∞ | 98.1 | 98.2 | 97.5 | 97.1 |

Table 10: Measuring the downstream performance on benign ($X$) and trigger-stamped ($\hat{X}$) after fine-tuning with AFMs backdoored with different layers' representations selected to inject the backdoor (i.e., when minimizing $\mathcal{L}_{Back}$). Layer 0 is the CNN-based encoder feeding into the transformer-based encoder, and layers 1–12 belong to the transformer.

| Task \ Layer | $\hat{X}$ 0 | 1 | 2 | 3 | 4 | 5 | 6 | 12 | $X$ 0 | 1 | 2 | 3 | 4 | 5 | 6 | 12 |
|---|---|---|---|---|---|---|---|---|---|---|---|---|---|---|---|---|
| KS↑ | 94.4 | 61.9 | 30.6 | 93.3 | 28.0 | 67.6 | 61.9 | 75.1 | 96.1 | 95.7 | 95.6 | 94.8 | 94.3 | 93.0 | 94.8 | 95.7 |
| ER↑ | 59.6 | 39.4 | 29.3 | 60.3 | 34.7 | 42.3 | 42.6 | 63.1 | 62.0 | 65.3 | 64.0 | 62.8 | 61.5 | 60.8 | 60.5 | 63.1 |
| ASR↓ | 14.3 | 63.7 | 98.2 | 14.0 | 98.4 | 93.9 | 96.7 | 96.2 | 11.3 | 11.5 | 11.7 | 11.6 | 11.4 | 11.6 | 11.5 | 12.4 |
| PR↓ | 8.1 | 92.5 | 99.7 | 10.7 | 99.8 | 99.9 | 99.9 | 99.0 | 5.4 | 5.4 | 5.6 | 5.8 | 5.4 | 5.6 | 5.5 | 5.4 |
| SID↑ | 80.4 | 35.8 | 34.9 | 62.6 | 0.7 | 2.5 | 70.7 | 74.3 | 81.7 | 33.5 | 32.8 | 65.5 | 77.0 | 74.0 | 71.3 | 76.1 |
| IC↑ | 96.4 | 23.3 | 7.2 | 92.6 | 6.6 | 12.0 | 7.7 | 22.0 | 98.4 | 98.4 | 98.1 | 97.9 | 98.2 | 95.1 | 97.9 | 98.3 |
| SD↓ | 6.8 | 13.9 | 21.9 | 7.5 | 25.3 | 9.3 | 9.0 | 8.3 | 6.2 | 6.4 | 6.4 | 7.0 | 6.6 | 6.7 | 6.5 | 6.3 |
| ASV↓ | 6.0 | 31.4 | 33.3 | 7.0 | 31.3 | 13.8 | 10.5 | 7.3 | 5.7 | 5.5 | 5.6 | 6.2 | 5.5 | 6.4 | 5.9 | 5.7 |
| ST↑ | 14.70 | 1.30 | 1.08 | 11.23 | 0.87 | 1.06 | 1.17 | 1.68 | 16.32 | 15.78 | 15.56 | 14.11 | 15.91 | 15.77 | 15.32 | 15.86 |

Table 11: The performance of the ASR downstream task (in WER↓) when fine-tuning models with AFMs backdoored using different losses as $\mathcal{L}_{Benign}$: Using MSE between the output representation and the original representation before AFM backdooring, or minimizing standard masked token-prediction loss (Eq. 2) with a codebook reproduced by the attack.

| $\mathcal{L}_{Benign}$ | $X$ | $\hat{X}$ |
|---|---|---|
| MSE | 14.2 | 59.4 |
| Masked token-prediction (Eq. 2) | 11.4 | 98.4 |

