# OpenReview forum: "Sounding the Alarm: Backdooring Acoustic Foundation Models for Physically Realizable Triggers"
_ICLR.cc/2025/Conference — Submitted to ICLR 2025_

### Official Review · Reviewer_zmPq · 2024-10-17

**Soundness:** 2
**Presentation:** 2
**Contribution:** 2
**Rating:** 3
**Confidence:** 5

**Summary:**

In this paper, the authors assume that the backdoor attacker performs a type of ‘man-in-the-middle’ attack between the (benign) model provider and the victim user. Specifically, the attacker injects a backdoor into a pretrained Acoustic Foundation Model (AFM) obtained from the provider, and then releases it to the victim, who further fine-tunes it for a downstream task. According to their claim, the proposed backdoor is physically realizable, inconspicuous, input-agnostic, and sync-free (not requiring synchronization between the trigger sound and the sonic input).

**Strengths:**

# Advantage
The main contribution of this work is the method for injecting the backdoor in a realistic scenario where the attacker has limited capability: the attacker can only access the pretrained model, cannot access the provider's training dataset, and has no knowledge of the downstream task. To inject a backdoor, the attacker reduces the representation distance between poisoned samples (benign sample + trigger) and a ‘non-useful’ representation from intermediate layers, without affecting the model’s performance on benign tasks. During the downstream task, once the trigger is present in the input, the input is mapped to the ‘non-useful’ representation, leading to misclassification.

The main challenge is maintaining the model’s performance on benign tasks. According to the authors, the best approach is to retrain the pretrained model in the same way the provider did (but with an additional backdoor loss). However, since the attacker does not know the exact codewords and other parameters, the authors propose approximating the missing parameters and using pseudo-labeling.

**Weaknesses:**

# Weakness

Several weaknesses can be identified:

1.	The authors assume that the user will use two fine-tuning paradigms: fine-tuning only the last layer or applying a weighted sum of all layers’ representations. However, to attach the trigger and the ‘non-useful’ representation, it seems necessary to freeze the feature extractor of the AFM. It is important to verify whether the backdoor remains robust after fine-tuning the entire AFM. Based on my experience, it is hard to ensure backdoor mapping without freezing.

2.	The authors also assume that the attacker knows the same training procedure as the provider. How realistic is it for the model to publicly release the training algorithm?

3.	The two defenses are outdated and specifically don’t focus on backdoor detection in SSL. The authors should also consider more recent methods, such as:1.	Zheng, Mengxin, et al. "SSL-Cleanse: Trojan Detection and Mitigation in Self-Supervised Learning." ECCV 2024;2.	Feng, Shiwei, et al. "Detecting Backdoors in Pre-Trained Encoders." Proceedings of the IEEE/CVF Conference on Computer Vision and Pattern Recognition. 2023.

**Questions:**

NA

---

> ### Author Response · Authors · 2024-11-27
>
> Thank you for the feedback!
>
> > Fine-tune mechanism
>
> We apologize for not explaining this point clearly enough. We tested two fine-tuning strategies: (1) For downstream tasks in the SUPERB benchmark, we fine-tuned only the downstream model (taking the weighted sum of all intermediate layers’ outputs as input) while keeping the AFM weights frozen; and (2) For the automated speech recognition (ASR), the upstream model was not frozen, and was updated simultaneously with the downstream model. We will clarify in the revision.
>
> > Defenses
>
> We thank the reviewer for suggesting defenses.
>
> After carefully reviewing the defenses, it seems that neither of these two methods is capable of defending against our attack. The defenses assume a backdoor that targets a specific class, while  our backdoor attack is conducted on an intermediate layer, targeting a predefined vector rather than a specific class. Additionally, the input features of the speech domain and the trigger stamping process differ significantly from those text and image domains. For instance, in the image domain, triggers are often added by replacing a patch in the image with the trigger, while, in our case, the trigger is overlayed over part of the audio. Altogether, these differences render the defenses inapplicable to the proposed attack.
>
>
> > Access to the training procedure
>
> We believe this might’ve been a misunderstanding.
>
> In prior works, the common assumption is that adversaries have full access to both the pretraining method and its associated modules, including the pretraining code, datasets [A, B, C, D, E]. In contrast, we do not assume that the adversary can access all training modules. Concretely, in the case of AFM, certain auxiliary modules used for pretraining are typically not open-sourced, including the codebook, projection matrix (upstream encoder), and pseudo-labels. Under the assumption that the adversary cannot directly access these auxiliary modules, we employ reverse engineering to regenerate the pseudo-labels and codebook, then implant the backdoor. Hence, our threat model is more realistic than prior work’s.
>
>
> **References**
>
> [A] Shen et al. "Backdoor Pre-trained Models Can Transfer to All." CCS, 2021.
>
> [B] Zhang et al. "Red alarm for pre-trained models: Universal vulnerability to neuron-level backdoor attacks." Machine Intelligence Research, 2023.
>
> [C] Jia et al. "BadEncoder: Backdoor attacks to pre-trained encoders in self-supervised learning." S&P, 2022.
>
> [D] Lyu et al. "Attention-enhancing backdoor attacks against BERT-based models." EMNLP Findings, 2023.
>
> [E] Xu et al. "Exploring the Universal Vulnerability of Prompt-based Learning Paradigm." NAACL Findings, 2022.

---

### Official Review · Reviewer_2nfy · 2024-10-31

**Soundness:** 3
**Presentation:** 3
**Contribution:** 3
**Rating:** 5
**Confidence:** 5

**Summary:**

This paper proposes a backdoor attack against the acoustic foundation model in self-supervised learning, which can degrade the performance of corresponding downstream tasks.

**Strengths:**

1. Language is mostly accessible, though with some minor issues
2. Experimental results verify the effectiveness of the proposed scheme.

**Weaknesses:**

The research motivation is not practical enough

**Questions:**

1. key concern: The author chose prominent sounds such as sirens, oboes, flutes, and bark as backdoor triggers, aiming to degrade the performance of associated downstream tasks rather than inducing a specific target behaviour. Theoretically, readers might question why attackers do not directly utilize noise-like triggers to impair the model performance if the attacker was required to activate the trigger actively. Besides, considering stealthiness, selecting white noise or ultrasound could obviously be seen as more covert than distinctive sounds like sirens. Therefore, It seems that the article's motivation is not clearly explained.
2. The methods section (Section 4) writing is overly redundant and tedious, making an otherwise straightforward approach appear complicated. It is recommended that the author revise the methods section to present the information more logically and streamlined.

---

> ### Author Response · Authors · 2024-11-27
>
> We thank the reviewer for the constructive  comments.
>
> > Motivation
>
> While noise-like triggers such as white noise or ultrasound can potentially be used for backdooring, they may be detected by anomaly detection or removed by (e.g., low-pass) filters. The triggers we opted to use (e.g., barks or sirens) are based on common sounds that are likely to blend into the background. Additionally, we find that the triggers barely affect the accuracy of clean models (Tables 1, 5–7, 9), thus rendering them inconspicuous.
>
> > Presentation
>
> We thank the reviewer for pointing out presentation issues. We will make Section 4 clearer.

---

### Official Review · Reviewer_xWgq · 2024-11-04

**Soundness:** 2
**Presentation:** 2
**Contribution:** 2
**Rating:** 3
**Confidence:** 5

**Summary:**

Foundation models enhance performance across various downstream tasks and minimize the need for labeled data. This paper's authors revealed that acoustic foundation models are vulnerable to backdoor attacks, and extensive experiments confirmed the effectiveness of these attacks.

**Strengths:**

Exploring the security issues in acoustic foundation models is both an important and intriguing topic.

**Weaknesses:**

1) The paper is poorly written.
2) The threat model lacks clarity.
3) Baseline attacks are not included.

**Questions:**

Thank you to the authors for this interesting paper. I have a few comments regarding the current work:

1) Overall, the paper is not well-written. In Section 4, instead of just stating that FAB minimizes a compound loss function, the authors should provide the mathematical details of  L_back and L_benign. Without this, it becomes difficult for readers to follow the explanation.
2) Typically, foundation models and downstream tasks use different encoders. However, the threat model does not clearly specify whether the attacker is aware of the encoders used in the downstream tasks.
3) In recent years, numerous papers have targeted foundation models to mislead downstream tasks, such as [A] and [B]. The main distinction in this paper seems to be its focus on the security of acoustic foundation models. Aside from this, the technical challenges appear similar. The authors should clarify in the related work section the key differences between this research and existing literature. Why not simply apply previous attack techniques to acoustic foundation models? What are the unique challenges in attacking acoustic foundation models?
4) The current paper does not seem to compare the proposed attacks with existing ones. It should not be difficult to apply the attacks described in [A] and [B] to the scenario outlined in this paper.
5) A potential defense against the proposed attack could be for the downstream task to use a conditional diffusion model to denoise the audio.


[A] Badencoder: Backdoor attacks to pre-trained encoders in self-supervised learning.

[B] Adversarial Illusions in Multi-Modal Embeddings.

---

> ### Author Response · Authors · 2024-11-27
>
> We would like to thank the reviewer for the constructive review.
>
> > Presentation
>
> Thank you for the suggestions for improving Section 4. We will revise the section to clarify the loss used in the attack (per the details already included in Appendix C.7).
>
> > Specification of encoders
>
> We don’t assume the attacker can access any information related to downstream models, including the encoders. Even under these strict constraints, we find that FAB is highly effective. We will clarify the assumption in the revision.
>
> > Baselines
>
> We thank the reviewer for pointing out potential baselines. After carefully reviewing the papers, we found they are unsuitable to serve as baselines due to having different goals and not adhering to our threat model’s assumptions: Jia et al. assume the attacker can access downstream task samples [A], and Bagdasaryan et al. produce input-specific adversarial perturbation [B].
>
> > Conditional diffusion
>
> We appreciate the suggestions. If the reviewer has a particular paper in mind, we would be glad to consider it. Please note that we already consider a defense to sanitize inputs and find it ineffective (Section 6.3).
>
> ## References
>
> [A] Jia et al. "BadEncoder: Backdoor attacks to pre-trained encoders in self-supervised learning." S&P, 2022.
>
> [B] Bagdasaryan et al. "Adversarial Illusions in Multi-Modal Embeddings." USENIX Security, 2024.

---

### Official Review · Reviewer_VWyN · 2024-11-04

**Soundness:** 3
**Presentation:** 3
**Contribution:** 1
**Rating:** 3
**Confidence:** 5

**Summary:**

This paper introduces the Foundation Acoustic model Backdoor (FAB) attack, a method for inserting backdoors in acoustic foundation models (AFMs) using physically realizable, input-agnostic audio triggers, such as sirens or barks. FAB attacks maintain model performance on benign data but degrade it significantly when activated, impacting various downstream tasks. Unlike prior work, FAB assumes a weak threat model, with attackers having limited knowledge of training data and model parameters. Experiments across multiple AFMs, tasks, triggers, and defenses show FAB's effectiveness in both digital and physical settings, even against standard defenses like fine-pruning and input filtering. This study highlights security risks in AFMs and the need for stronger defenses against backdoor attacks.

**Strengths:**

- Task-Agnostic and Physically Realizable Backdoor Attack: The proposed FAB is task-agnostic and physically realizable. Unlike prior work, FAB uses simple, inconspicuous sounds like sirens or dog barks as triggers without needing synchronization, making it adaptable to real-world settings.

- Comprehensive Evaluation: Extensive experiments demonstrate FAB's effectiveness across various tasks and AFMs. The paper also tests FAB against defenses like fine-pruning and input filtration, showing its resilience. This thorough evaluation highlights the attack’s robustness and underscores the need for stronger defenses in model security.

**Weaknesses:**

- Limited Novelty: While the FAB attack presents an interesting application, fine-tuning-based backdoor injection is not new, and FAB still requires auxiliary data with a distribution similar to the target AFM. This reliance on auxiliary data limits its originality and does not fully address existing challenges in realistic backdoor attacks.

- Unrealistic Threat Model: The threat model lacks practical relevance, as it assumes the model provider itself would inject backdoors that degrade performance significantly. Major AFM providers, like OpenAI, are unlikely to introduce such vulnerabilities into their own models, limiting the real-world applicability of this attack.

- Limited Evaluation Scope: The study focuses primarily on AFMs like HuBERT and WavLM, without assessing generalizability to a broader range of models. Testing additional AFM architectures, such as wav2vec 2.0 from Google and Data2Vec from Meta.

- Lack of Audio Quality Evaluation: The study does not assess the audio quality of samples embedded with backdoor triggers compared to the original, unmodified samples. Metrics like ViSQOL could provide insights into whether the trigger significantly affects perceived audio quality. This evaluation is essential to ensure that the trigger remains inconspicuous to users, as noticeable quality degradation could alert users to the presence of an anomaly, undermining the stealth of the attack.

**Questions:**

- Could you elaborate on the need for auxiliary data with a similar distribution to the target AFM? Are there any approaches under consideration to remove this requirement? Clarifying this could help address concerns regarding FAB's novelty and broader applicability.

- Given that major AFM providers like OpenAI are unlikely to risk their model’s reputation by embedding backdoors, who do you envision as the realistic adversary for FAB? Additional clarification on feasible attacker scenarios could strengthen the practical relevance of the proposed method.

- The study’s evaluation primarily focuses on HuBERT and WavLM. Have you considered expanding your experiments to include other prominent AFMs, such as wav2vec 2.0 or Data2Vec?

- To better understand the stealth of the embedded trigger, could you provide an audio quality evaluation using metrics like ViSQOL?

---

> ### Author Response · Authors · 2024-11-27
>
> We thank the reviewer for the valuable feedback.
>
> > Novelty
>
> To the best of our knowledge, our work is the first to _backdoor speech foundation models through inconspicuous, physically realizable, input-agnostic, and sync-free attacks that are agnostic to the downstream task_. Our attack also does not use pre-training or task-specific data, and instead _relies on readily available auxiliary data_ for backdooring. In comparison, prior work either assumed adversary access to pre-training data (e.g., [A, B, C, D, E]) or downstream data (e.g., [C, F, G, H, I]), thus rendering established attacks less realistic.
>
> > Realism of the threat model
>
> We don't assume model providers introduce the backdoors themselves. Our adversary starts from a well-trained open-source model (e.g., one published on HuggingFace) and creates a counterpart with similar benign performance but with a backdoor injected. The adversary can then publish its seemingly benign and well-performing model on a public repository or provide it as part of a supply-chain attack.
>
> > Evaluation scope
>
> Since HuBERT and WavLM are two of the most competitive AFMs per standard benchmarks (e.g., SUPERB), we believe they convincingly demonstrate the effectiveness and generality of the proposed attacks. Because the AFM proposed by the reviewer has a similar architecture to the two we already test, we don’t expect that experimenting with it would change the overall takeaways.
>
> > Audio quality evaluation
>
> To demonstrate that trigger-stamped (i.e., poisonous) inputs are inconspicuous, we vary the signal-to-noise ratio (Table 9) and show that benign AFMs (without backdoor) remain accurate on these inputs (Tables 1, 5–7, 9). These results show that poisonous samples still preserve their semantic meaning after introducing triggers.
>
> **References**
>
> [A] Shen et al. "Backdoor Pre-trained Models Can Transfer to All." CCS, 2021.
>
> [B] Zhang et al. "Red alarm for pre-trained models: Universal vulnerability to neuron-level backdoor attacks." Machine Intelligence Research, 2023.
>
> [C] Jia et al. "BadEncoder: Backdoor attacks to pre-trained encoders in self-supervised learning." S&P, 2022.
>
> [D] Lyu et al. "Attention-enhancing backdoor attacks against BERT-based models." EMNLP Findings, 2023.
>
> [E] Xu et al. "Exploring the Universal Vulnerability of Prompt-based Learning Paradigm." NAACL Findings, 2022.
>
> [F] Kurita et al. "Weight poisoning attacks on pre-trained models." ACL, 2020.
>
> [G] Yang et al. "Be careful about poisoned word embeddings: Exploring the vulnerability of the embedding layers in NLP models." NAACL, 2021.
>
> [H] Yang et al. "Rethinking stealthiness of backdoor attack against NLP models." ACL. 2021.
>
> [I] Li et al. "Backdoor attacks on pre-trained models by layerwise weight poisoning." EMNLP, 2021.

---

### Meta-Review · Area_Chair_yBtW · 2024-12-18

**Metareview:**

The recommendation is based on the reviewers' comments, the area chair's evaluation, and the author-reviewer discussion.

While the reviewers see some merits in exploring the backdoor risks in acoustic foundation models,
this submission should not be accepted in its current form due to several fundamental issues, as pointed out by the reviewers, including

- Limited novelty
- Unrealistic threat model
- Insufficient evaluation

During the final discussion phase, all reviewers agreed to reject this submission. The authors' rebuttal were not able to alter the reviewers' assessment.

I hope the reviewers’ comments can help the authors prepare a better version of this submission.

**Additional Comments On Reviewer Discussion:**

This submission should not be accepted in its current form due to several fundamental issues, as pointed out by the reviewers, including

- Limited novelty
- Unrealistic threat model
- Insufficient evaluation

During the final discussion phase, all reviewers agreed to reject this submission. The authors' rebuttal were not able to alter the reviewers' assessment.

---

### Decision · Program_Chairs · 2025-01-22

Reject